# Identification of Muscle Strength-Related Gut Microbes through Human Fecal Microbiome Transplantation

**DOI:** 10.3390/ijms25010662

**Published:** 2024-01-04

**Authors:** Ji-Seon Ahn, Bon-Chul Koo, Yu-Jin Choi, Woon-Won Jung, Hyun-Sook Kim, Suk-Jun Lee, Seong-Tshool Hong, Hea-Jong Chung

**Affiliations:** 1Gwangju Center, Korea Basic Science Institute, Gwangju 61751, Jeolla, Republic of Korea; ajs0105@kbsi.re.kr (J.-S.A.); cyj4854@kbsi.re.kr (Y.-J.C.); 2Department of Biomedical Sciences and Institute for Medical Science, Jeonbuk National University Medical School, Jeonju 54907, Jeonbuk, Republic of Korea; 3Division of Bioconvergence Analysis, Korea Basic Science Institute, Ochang 28119, Chungbuk, Republic of Korea; koobc@kbsi.re.kr; 4Department of Biomedical Laboratory Science, Cheongju University, Cheongju 28503, Chungbuk, Republic of Korea; woonun@cju.ac.kr (W.-W.J.); hskim-97@cju.ac.kr (H.-S.K.); juna332@hanmail.net (S.-J.L.)

**Keywords:** muscle strength, gut microbiome, fecal microbiome transplantation, *Eisenbergiella massiliensis*, *Anaeroplasma abactoclasticum*, *Ileibacterium valens*, *Ethanoligenens harbinense*

## Abstract

The gut microbiome is well known for its influence on human physiology and aging. Therefore, we speculate that the gut microbiome may affect muscle strength in the same way as the host’s own genes. To demonstrate candidates for gut microbes affecting muscle strength, we remodeled the original gut microbiome of mice into human intestinal microbiome through fecal microbiome transplantation (FMT), using human feces and compared the changes in muscle strength in the same mice before and three months after FMT. After comparing before and after FMT, the mice were divided into three groups based on the observed changes in muscle strength: positive, none, and negative changes in muscle strength. As a result of analyzing the α-diversity, β-diversity, and co-occurrence network of the intestinal microbial community before and after FMT, it was observed that a more diverse intestinal microbial community was established after FMT in all groups. In particular, the group with increased muscle strength had more gut microbiome species and communities than the other groups. Fold-change comparison showed that *Eisenbergiella massiliensis* and *Anaeroplasma abactoclasticum* from the gut microbiome had positive contributions to muscle strength, while *Ileibacterium valens* and *Ethanoligenens harbinense* had negative effects. This study identifies candidates for the gut microbiome that contribute positively and those that contribute negatively to muscle strength.

## 1. Introduction

In general, humans have a complex and massive microbial community consisting of 100 trillion microbes in the intestine, called the gut microbiome [1]. Recent studies have shown that the gut microbiome plays an important role in overall health, including digestion, immunity, and even mental health [2,3,4,5]. Gut microbial imbalance causes a variety of diseases, including inflammatory bowel disease, gastrointestinal malignancies, cancer, cholelithiasis, autism, sarcopenia, cachexia, hepatic encephalopathy, allergy, obesity, diabetes, atherosclerosis, metabolic syndrome, Alzheimer’s disease, Parkinson’s disease, etc. [6,7,8,9,10,11,12,13,14,15]. Although current research clearly shows that the gut microbiome influences the host’s digestive, immune, and mental health [16], the impact of the gut microbiome on physiology has not been thoroughly investigated.

Considering the results of current research on the function of the gut microbiome, there is a strong possibility that the gut microbiome may affect the muscle strength of its host as much as the host’s own genes do [17]. However, the role of the gut microbiome in muscle strength has shown mixed results. Initial studies did not show a positive role of the gut microbiome in the maintenance of whole-body lean mass [16]. Whole-body lean mass was decreased by 7–9% in young germ-free mice after transplantation of fecal samples from conventionally raised mice [16]. In contrast to the initial research, subsequent research showed that the gut microbiome increased skeletal muscle mass, reduced muscle atrophy markers, improved the oxidative metabolic capacity of the muscle, and elevated expression of the neuromuscular junction assembly genes *Rapsyn* and *Lrp4* [18,19,20]. Furthermore, transplantation of the gut microbiome from young mice to old mice has been shown to increase muscle fiber thickness and grip strength [21].

There is some evidence to suggest that the gut microbiome may be associated with muscle function. This evidence suggests that the gut microbiota composition and diversity can be a determinant of skeletal muscle metabolism and functionality. [22] Longitudinal studies have explored the connection between a physically active lifestyle or long-term exercise interventions and the gut microbiota [23].

Additionally, studies in mice have shown that altering the composition of the gut microbiome can affect motor behavior [24,25]. Studies in humans have also suggested a link between the gut microbiome and physical activity [26,27].

Although these studies suggested a possible association between the gut microbiome and muscle strength, there is no direct evidence linking the gut microbiome to muscle strength. Mammalian phenotypes, such as muscle strength, are influenced by both genes and the gut microbiome [22,23,24,25,26,27,28,29,30]. Genetic variation among individuals means that evaluating the effect of the gut microbiome on muscle strength is greatly influenced by the genetic background of the host. In particular, human studies, in which individual genetic backgrounds cannot be controlled, have limitations in accurately evaluating the effects of the gut microbiome. In contrast, mouse studies, where genetic factors can be controlled, allow for a more accurate evaluation of the effects of the gut microbiome by eliminating the influence of genetic factors [31].

When evaluating the effect of the gut microbiome on muscle strength, it is difficult to be sure whether the results are due to genetic factors or the influence of the gut microbiome. To overcome these difficulties, this study developed a new gut microbiome analysis method. This method reconstructed the gut microbiome through fecal microbiome transplantation (FMT) under identical environmental conditions and then compared the muscle strength of the same individual mice before and after FMT, eliminating the genetic background. This method more accurately determined the influence of gut microbes alone on muscle strength, excluding genetic and environmental factors.

## 2. Results

### 2.1. Randomly Colonizing Conventional Mice with Human Gut Microbiome via FMT Had Differential Effects on Muscle Strength

The ongoing debate regarding the regulation of muscle strength by intestinal bacteria prompted us to explore a new approach to identify the specific bacteria responsible. We performed a novel in vivo experiment utilizing the concept of subgroup analysis, which can help find the causes of complex problems in large datasets. This study aimed to identify the intestinal bacteria that regulate muscle strength through randomized subgroup analysis. We depleted the gut microbiome of conventional mice (C57BL/6) with a mixture of three broad-spectrum antibiotics and antifungal and randomly colonized human gut microbiome by feeding them fresh fecal samples (Figure 1A).

The effect of FMT on muscle strength in conventional mice was vastly different among mice, as shown in Figure 1B,C, probably due to the human gut microbiota randomly colonizing the mice gut. To rule out genetic factors and evaluate the impact of the gut microbiome alone on muscle strength changes, we replaced the original gut microbiome with the human gut microbiome via FMT and then monitored the rotarod records in the same mice. Changes in muscle strength in mice over the three-month experimental period can be classified into three categories: the group with increased muscle strength, in which the holding time on the rotarod increased by 23.5 ± 5.7 s (Strong Muscle Strength; SMS; *n* = 10); the group with unchanged muscle strength, in which the holding time on the rotarod remained within the range of 1.6 ± 0.8 s (Medium Muscle Strength; MMS; *n* = 10); and the group with decreased muscle strength, in which the holding time on the rotarod decreased by −16.1 ± 3.2 s (Weak Muscle Strength; WMS; *n* = 10) (Figure 1B,C). Muscle strength changes in each group were confirmed through histological examination, which revealed high levels of muscle fiber accumulation in the SMS, intermediate accumulation in the MMS, and the lowest accumulation in the WMS (Figure 1D). The blood glucose levels and lipid profiles did not change significantly either before or after FMT in mice (Appendix A). These results suggest that different subsets of the human gut microbiome randomly replaced the original gut microbiome in each mouse, resulting in different effects on muscle strength.

### 2.2. After Performing FMT with Human Feces, Different Types of Gut Microbial Communities Were Established in Each Experimental Mouse

The diverse effects on muscle strength following FMT with human feces led us to compare changes in the gut microbiota composition before and after FMT in mice. We sequenced the V3–V4 regions of the 16S rRNA genes of the gut microbiome of each mouse before and after replacement using the MiSeq platform (Illumina). We filtered out sequence reads that potentially contained incorrect primer or barcode sequences, sequences with more than one ambiguous base, low-quality sequences, or chimeras, which comprised approximately 0.001% of the total reads. The filtered 16S rRNA sequences were used to identify individual microbes by matching them with the SILVA reference database (region V3–V4) (https://www.arb-silva.de/ (accessed on 20 July 2021)). The identified 16S rRNA sequences were classified into nine phyla: Bacteroidetes (48.471%), Firmicutes (37.456%), Verrucomicrobia (10.197%), Proteobacteria (1.924%), Patescibacteria (0.337%), Actinobacteria (0.907%), Tenericutes (0.48%), Cyanobacteria (0.228%), and Lentisphaerae (0.002%) (Figure 2 and Appendix A). The operational taxonomic units (OTUs) were further classified down to species level.

Comparisons of OTUs showed clear differences before and after FMT in all mice, indicating the success of replacing the original gut microbiome with the human microbiome, as shown in Figure 2. The composition of the replaced gut microbiome was very different among the groups, classified by individual differences in muscle strength before and after FMT (Figure 3 and Appendix A, Appendix A). Before FMT, the main bacteria that constituted the original gut microbiomes at the phylum level were Bacteroidetes (56.093% in SMS, 53.922% in MMS, and 63.633% in WMS), Firmicutes (40.116% in SMS, 42.569% in MMS, and 33.938% in WMS), and Patescibacteria (3.24% in SMS, 2.791% in MMS, and 2.03% in WMS); at the class level, Bacteroidia (56.093% in SMS, 53.922% in MMS, and 63.633% in WMS), Clostridia (36.684% in SMS, 38.674% in MMS, and 29.873% in WMS), and Bacilli (2.489% in SMS, 2.499% in MMS, and 3.37% in WMS); at the order level, Bacteroidales (56.093% in SMS, 53.92% in MMS, and 63.628% in WMS), Clostridiales (36.684% in SMS, 38.674% in MMS, and 29.873% in WMS), and Lactobacillales (2.489% in SMS, 2.499% in MMS, and 3.37% in WMS); and at the family level, Muribaculaceae (54.598% in SMS, 50.66% in MMS, and 61.67% in WMS), Lachnospiraceae (24.471% in SMS, 24.611% in MMS, and 19.667% in WMS), and Ruminococcaceae (9.72% in SMS, 10.737% in MMS, and 7.399% in WMS). However, the composition of bacteria changed three months after FMT: at the phylum level, Bacteroidetes (52.068% in SMS, 43.928% in MMS, and 49.416% in WMS), Firmicutes (36.244% in SMS, 37.628% in MMS, and 38.496% in WMS), and Verrucomicrobia (7.552% in SMS, 14.915% in MMS, and 8.123% in WMS); at the class level, Bacteroidia (52.068% in SMS, 43.928% in MMS, and 49.416% in WMS), Clostridia (30.437% in SMS, 32.948% in MMS, and 32. 293% in WMS), and Verrucomicrobiae (7.552% in SMS, 14.915% in MMS, and 8.123% in WMS); at the order level, Bacteroidales (56.093% in SMS, 53.92% in MMS, and 63.628% in WMS), Clostridiales (36.684% in SMS, 38.674% in MMS, and 29.873% in WMS), and Lactobacillales (2.489% in SMS, 2.499% in MMS, and 3.37% in WMS); at a family level, Muribaculaceae (40.613% in SMS, 33.303% in MMS, and 40.784% in WMS), Lachnospiraceae (16.383% in SMS, 18.101% in MMS, and 19.08% in WMS), and Ruminococcaceae (12.561% in SMS, 13.346% in MMS, and 11.583% in WMS) (Appendix A). These results indicate that the human gut microbiome is not only more diverse but also that its composition significantly differs from that of mice.

### 2.3. Changes in Muscle Strength Were Correlated with Changes in the Gut Microbiome Composition

After confirming the correlation between individual differences in the composition of the replaced gut microbiome and muscle strength, the diversity of the gut microbiome of each mouse was analyzed. The α-diversity metrics showed that different subsets of the human gut microbiome were replaced in each group of mice (Figure 4 and Appendix A). The α-diversity indices, which take into account both richness (ACE and Fisher’s alpha) and evenness (Shannon, Simpson, and InvSimpson), showed little difference among the three groups. Comparing before and after FMT, the evenness indices before and after FMT were similar, but the richness indices increased after FMT, suggesting that although the compositional characteristics of the human and mouse gut microbiomes are similar, the human gut microbiome is more diverse. Additionally, the richness and evenness diversity indices of mice with stronger muscle strength (SMS) were higher than those of the other two groups, indicating that the gut microbiome was more diverse in SMS.

In addition to α-diversity analyses, β-diversity analyses also confirmed that the replaced gut microbiome was significantly more diverse than the original gut microbiome of each mouse. Both β-diversity metrics measured via non-metric multidimensional scaling (NMDS), and principal coordinate analysis (PCoA) plots showed that the compositions were more diverse after FMT (Figure 5A,B and Appendix A).

To investigate the effect of the gut microbiome on muscle strength, we conducted a comparison involving unsupervised hierarchical clustering of the most abundant operational taxonomic units (OTUs). This comparison was based on the Bray–Curtis distance of the gut microbiome, along with assessments of changes in muscle strength before FMT (T0) and three months after FMT (T3). It was shown that gut microbiome replacement via FMT affected the three groups of mice differently. An unsupervised hierarchical cluster analysis before FMT (T0) revealed no differences between each mouse group (Figure 5C, left). However, a hierarchical cluster analysis clearly revealed differences across groups in muscle strength changes after FMT (Figure 5C, right). These showed the gut microbiome composition, grouping individual mice according to differences in muscle strength.

### 2.4. Different Microbial Communities Were Established in Each of the Three Groups of Mice after FMT

Microbial compositions showed that the replaced gut microbiomes of the three groups following muscle strength changes were different, even though the original gut microbiomes were not different from each other (Figure 3). Similarly, a co-occurrence network analysis showed that the microbial communities became more diverse after FMT in all three groups (Figure 6 and Appendix A). Before FMT, 21 communities were identified in SMS, 28 in MMS, and 31 in WMS, but after FMT, the communities expanded to 74 in SMS, 60 in MMS, and 58 in WMS. The number of nodes and edges within the microbial communities were significantly increased in all three groups before and after FMT. We found that the number of microbial communities was higher in SMS than in MMS and WMS after FMT, which suggests that muscle strength is associated with a more diverse gut microbiome.

### 2.5. Gut Microbes That Influence Muscle Strength Were Identified at the Species Level

Although the above gut microbiome analyses revealed a clear correlation between the gut microbiome composition and muscle strength, it did not identify the specific bacterial groups responsible for either promoting or reducing muscle strength (Figure 3, Figure 4, Figure 5 and Figure 6). We utilized the concept of fold change at the log2 scale and a linear correlation to analyze the abundance of intestinal microbes in relation to muscle strength (Figure 7). Among the bacterial species, *Eisenbergiella massiliensis* and *Anaeroplasma abactoclasticum* were the most abundant in the SMS, indicating a positive correlation with muscle strength. In contrast, *Ileibacterium valens* and *Ethanoligenens harbinense* were the most abundant in WMS, indicating a negative correlation with muscle strength. Bacteria associated with strong muscle strength were classified into the phyla Firmicutes and Tenericutes, whereas those associated with weak muscle strength belonged to the phylum Firmicutes.

## 3. Discussion

It is well known that the gut microbiome plays an important role in the host’s energy extraction from food and increases villous blood vessel formation [32,33]. Moreover, recent studies have shown that the gut microbiome has a significant impact on various physiological processes, including metabolism, digestion, immunity, and brain function [2,3,4,5]. In addition, the gut microbiome acts as an important epigenetic factor regulating host gene expression [34,35,36,37,38].

Considering the diverse roles of the gut microbiome, it is expected to also play an important role in determining muscle strength in the host. However, previous studies have not shown a clear link between the gut microbiome and muscle strength. In contrast to the initial study that found a negative association between the gut microbiome and muscle strength [16], later studies have shown that the gut microbiome can actually enhance muscle strength [18,19,20]. The gut microbiome is a complex ecosystem composed of hundreds or thousands of species of bacteria and fungi [39]. The number of microorganisms in the gut microbiome can varies from person to person. Considering the diversity of each individual’s gut microbiome and the vast amount of gut microorganisms, the composition of the gut microbiome can manifest in many different combinations. That is, some gut microbiomes may have a negative effect on muscle strength while others may have a positive effect [40,41]. As a result, it is not surprising to see mixed evidence for the association between muscle strength and the gut microbiota in previous studies. In terms of bacterial species that may positively impact muscle mass, oral gavage with *Lactobacillus casei* or *Bifidobacterium longum* increased the muscle mass/body weight ratio without affecting body weight [42]. Recent findings from another research group contribute to the elucidation of the gut–muscle axis in older adults. This group identified higher levels of *Prevotellaceae*, *Prevotella*, *Barnesiella*, and *Barnesiella intestinihominis* in older adults in conjunction with higher muscle strength (defined as high-functioning, HF), when compared with older adults that had reduced muscle strength (defined as low-functioning, LF) [43].

There are uncontrolled limitations to using metagenomics to compare gut microbiomes between control and experimental groups. To overcome these limitations, a new method for gut microbiome analysis was developed in this study. This method involves transplanting the human fecal microbiomes into mice and observing phenotypic changes in the same individual mice. Subset analysis is a very useful method for finding subtle differences that are not found in a large dataset. Therefore, we took a valuable approach by transplanting the human gut microbiome (large dataset) into mice and subsequently analyzing the transplanted microbiome in mice (subset data). Additionally, by comparing phenotypic changes, such as changes in muscle strength, within the same individual mouse while controlling for genetic factors that influence muscle strength, we can reveal the unique influence of the gut microbiome on phenotypes [31].

Our results showed that the gut microbiome has a dual effect on muscle strength: some gut microbiomes have a positive effect on muscle strength, while others have a negative effect (Figure 1). Alpha and beta diversity, phylogenetic analyses and co-occurrence network assessments have confirmed that the gut microbial community differs in terms of the composition and diversity of gut microorganisms, affecting changes in muscle strength (Figure 2, Figure 3, Figure 4, Figure 5 and Figure 6). These findings indicated that the gut microbiome that influence muscle strength differs between groups. Our findings demonstrate that the gut microbiome contributes either positively or negatively to muscle strength, which clearly explains the mixed results of previous research on the association between muscle strength and the gut microbiome [18,19,20,21].

Additionally, through FMT-based gut microbiome subset analysis, we also identified specific gut microbes that are either positively or negatively associated with muscle strength (Figure 7). As depicted in Figure 7, *E. massiliensis* and *A. abactoclasticum* had a positive effect on muscle strength, whereas *I. valens* and *E. harbinense* had a negative impact. *A. abactoclasticum* is an obligate anaerobic bacterium [44]. Although *A. abactoclasticum* has been isolated from healthy chickens and cattle, they have not yet been found in humans. *E. massiliensis* is a gram-negative, obligate anaerobic bacterium that was isolated from the stool of an overweight person [45]. Notably, the two bacteria that had a negative effect on muscle strength, *I. valens* and *E. harbinense*, were found in lean individuals [46,47]. This finding of bacteria being positively associated with muscle strength in overweight individuals and vice versa in lean individuals seems to align well with human physiology.

As a result, the gut microbiome remodeled through FMT in each group also differed. Fold-change comparison showed that *Eisenbergiella massiliensis* and *Anaeroplasma abactoclasticum* from the gut microbiome had positive contributions to muscle strength, while *Ileibacterium valens* and *Ethanoligenens harbinense* had negative effects. Therefore, this study successfully demonstrated that in vivo subset analysis of the human gut microbiome using animal models is a useful approach for studying complex and heterogeneous populations comprised of trillions of microorganisms, and we believe that this concept can be applied to identify gut microbiome associated with other human phenotypes or diseases [48].

## 4. Materials and Methods

### 4.1. Study Design and Animal Experiments

Thirty 47-week-old C57BL/6 mice were purchased from the Animal Facility of Aging Science at the Korea Basic Science Institute (Gwangju, Republic of Korea) and acclimated for one week. The mice were housed individually to avoid exposure to feces from other mice. The mice were maintained under a specific light/dark cycle and temperature and had access to sterile food and water in a sterile environment. After one week of acclimation, the mice were weighed, feces and blood collected, and rotarod performed for pre-FMT recordings. To deplete their endogenous gut microbiota, the mice were treated the water containing antibiotics (1 g/L ampicillin, 0.5 g/L kanamycin, and 0.5 g/L cefoxitin; Sigma-Aldrich, St. Louis, MO, USA) and an antifungal (0.5 g/L nystatin) for one week.

The optimized media for collecting human fecal samples were prepared as previously described [49]. To collect diverse human gut microorganisms, fecal samples from 10 healthy volunteers were collected and mixed in the optimized media. The mixed fecal media were prepared on the scheduled day for oral gavage and stored at room temperature in anaerobic containers to preserve as many microorganisms as possible. This mixed human fecal media were used to perform FMT in mice. FMT was performed twice a week for a total of three months by administering 20 μL of the mixture to the mice via oral gavage. After 3 months of FMT, all mice were weighed, feces and blood were collected, and rotarod performed individually for post-FMT recording.

The changes in the muscle strength of the mice over the three-month experimental period can be grouped into three categories: the group with increased muscle strength (Strong Muscle Strength; SMS; n = 10); the group with unchanged muscle strength (Medium Muscle Strength; MMS; n = 10); and the group with decreased muscle strength (Weak Muscle Strength; WMS; n = 10).

### 4.2. Rotarod Test

The rotarod apparatus (B.S Technolab INC, Seoul, Republic of Korea) was used to evaluate the motor coordination, strength, and balance of the mice [50]. The apparatus consisted of a base platform and a rotating rod with a diameter of 3.5 cm and a non-slippery surface. The rod was placed at a height of 30 cm above the base. The mice were placed on the rotating rod, which was gradually increased in speed, and the time it took for the mouse to fall off was measured. The mice underwent three trials per day at 30 rpm for three consecutive days. The mice were tested on the fourth day. During testing, each animal underwent three trials at a fixed speed of 30 rpm. The mean latency to fall off the rotarod was recorded and used in the subsequent analyses.

### 4.3. Biochemical Parameter Analysis

Blood glucose levels were measured using a portable blood glucose meter (Accu-Chek Active; Roche Diagnostic GmbH, Mannheim, Germany). Total cholesterol (TCHO), triglyceride (TG), and high-density lipoprotein cholesterol (HDL-CHO) levels in mice serum were measured enzymatically using commercial assay kits (Asan Pharmaceutical Co., Seoul, Republic of Korea), as previously described [49]. Additionally, the low-density lipoprotein cholesterol (LDL-CHO) levels were calculated using the Friedewald’s equation [(LDL-CHO) = (TCHO) − ((HDL-CHO) − (TG)/5)].

### 4.4. Histological Analysis

The histological analysis was performed as described previously [49]. In brief, at the end of the experiment, muscle tissues were prepared from mice, fixed in 10% neutral buffered formalin, embedded in paraffin. After serial 6 µm thick sections, tissue sections were deparaffinized using hot water and stained with hematoxylin and eosin (Vector Laboratories Inc., Newark, CA, USA). H and E-stained tissue sections were observed under a light microscope (AmScope, T690C-PL, Irvine, CA, USA), and images were taken with a microscopic digital camera (AmScope, MU-1803, Irvine, CA, USA).

### 4.5. DNA Extraction and 16S rRNA Gene Sequencing

Total bacterial genomic DNA from fecal samples from each mouse was extracted using the phenol-chloroform isoamyl alcohol extraction method, as described previously [51]. In brief, fecal samples were suspended in lysis buffer (200 mM NaCl, 200 mM Tris-HCl (pH 8.0), 20 mM EDTA) via bead-beating. Genomic DNA was isolated through successive phenol:chloroform:isoamyl alcohol, 3M sodium acetate, and isopropanol treatments, washed with 70% ethanol, and dissolved in TE buffer (10 mM Tris-HCl (pH 8.0) and 1 mM EDTA). The concentration and purity of the extracted DNA were measured using a BioSpec-nano spectrophotometer (Shimadzu Biotech, Kyoto, Japan), and the integrity was assessed on a 1% (*w*/*v*) agarose gel.

16S rRNA gene sequencing analyses of the gut microbiome DNA samples were performed by a commercial company (ebiogen, Inc., Seoul, Republic of Korea). Briefly, each sample was prepared according to the Illumina 16S rRNA gene sequencing library protocol, and the genes were amplified using 16S V3–V4 primers; 16S Amplicon PCR Forward Primer with a sequence of 5′-TCGTCGGCAGCGTCAGATGTGTATAAGAGACAGCCTACGGGNGGCWGCAG-3′ and 16S Amplicon PCR Reverse Primer with a sequence of 5′-GTCTCGTGGGCTCGGAGATGTGTATAAGAGACAGGACTACHVGGGTATCTAATCC-3′. Subsequent limited-cycle amplification steps were performed to add the multiplexing indices and Illumina sequencing adapters. The final products were normalized and pooled using PicoGreen, and the size of the libraries was verified using the Agilent TapeStation DNA ScreenTape D1000 system (Agilent Technologies, Santa Clara, CA, USA). Finally, the pooled libraries (2 × 300) were sequenced using the MiSeq platform (Illumina, San Diego, CA, USA). The amplicon errors were modelled in merged fastq using DADA2 (ver.1.10.1). Noise sequences were filtered, errors in marginal sequences were corrected, chimeric sequences and singletons were removed, and sequences were de-duplicated [52]. The taxonomy profile data were deposited in the repository at figshare (https://doi.org/10.6084/m9.figshare.21838947) accessed on 9 January 2023.

### 4.6. Data and Statistical Analyses

All Data and statistical analyses were performed as described previously [51]. Briefly, to classify bacterial species, the Q2-Feature classifier that was trained based on the SILVA reference (region V3–V4) database (https://www.arb-silva.de/ (accessed on 20 July 2021)) after setting the de-noise-single function as the default parameter.

The q2-diversity, with the sampling depth option, was used for diversity calculations and statistical tests. We used a minimum sequencing quality score threshold of 20 and a rarefaction depth of 11,510. After checking the quality of the sequencing results, the sequencing results in the “table.qza” file were filtered using the threshold in QIIME 2. The metagenomic data OTUs and taxonomic classification tables were imported into the phyloseq (1.28.0) package in R version 3.6.1 for visualization of alpha and beta diversity. Statistical analysis was performed using the Kruskal–Wallis rank-sum test for alpha diversity. To detect statistical differences in beta diversity metrics between groups, we used permutational multivariate analysis of variance (PERMANOVA) in the vegan package in R. ADONIS was used with 999 permutations in the vegan package in R to quantify the effect size of variables explaining the Bray–Curtis distance. All *p*-values were corrected using Benjamini and Hochberg’s adjustment, and significance was declared at *p* < 0.05.

### 4.7. The α-, β-Diversity Analyses and Relative Abundance Evaluation of Gut Microbiome

The α-diversity analysis was performed as previously described [51]. We used the phyloseq (1.28.0) and metagenomeSeq (1.16.0) packages to identify the central taxa present in each group, then metadata, OTUs, and taxonomic classification tables were imported into the phyloseq package, and the data were processed according to the instructions [53,54]. Subsequently, the phyloseq class objects were converted to metagenomeseq objects and normalized using cumulative sum scaling (CSS) built specifically for metagenomic data in the BioConductor package metagenomeSeq (1.16.0). For further analysis and visualization, normalized data were converted to phyloseq class objects in R.

Normalized OTU data were used for abundance calculations and each taxonomic level was calculated for plotting. To clearly visualize abundance data, taxa were collected as “other” if their relative abundance was less than 5%, excluding phylum and class levels.

The β-diversity was computed for non-metric multidimensional scaling (NMDS) and multidimensional scaling (MDS) from log-transformed OTU data using Bray–Curtis dissimilarity in the vegan package, as described previously [51].

### 4.8. Construction of Heatmap

Heatmaps and cluster analyses were generated using the relative abundances of all OTU values in the Heatplus (2.30.0) package from Bioconductor and the vegan package in R, as described previously [31,55]. Average linkage hierarchical clustering and Bray–Curtis distance metrics were used for cluster analysis and heatmap generation, respectively. To collect the most abundant taxa for heatmap generation, we performed unsupervised prevalence filtering with a 5% threshold on all samples.

### 4.9. Co-Occurrence Network Construction

To observe the microbial co-occurrence relationships through muscle strength changes, co-occurrence networks were created using a permutation-renormalization-bootstrap network construction strategy, as described previously [31,55]. First, unnormalized abundance data were uploaded to CoNet, a Java Cytoscape plug-in. Microbial networks and links or edges were obtained from OTU occurrence data. CoNet’s multiple ensemble correlation method was used to identify significant co-presence across the samples, while OTUs occurring in less than three samples were discarded (“row_minocc” = 3). To create the ensemble network, we used CoNet to calculate five similarity measures, including Spearman and Pearson correlation coefficients, the Mutual Information Score, and the Bray–Curtis and Kullback–Leibler Dissimilarity; the *p*-value was merged using Brown’s method. The *p*-value was corrected using the Benjamini–Hochberg correction method (adjusted *p*-value < 0.05). If at least two of the five metrics suggested a significant co-existence between two OTUs, that relationship was maintained in the final network and is represented as an edge. The final co-occurrence network model was displayed using the igraph package in R by implementing the Louvain algorithm to identify communities within each network so that the modularity score of each OTU was maximized within a given network.

### 4.10. Differential Abundance

To estimate fold-changes of taxa in the gut microbiome according to the muscle strength change group, we used DESeq2 (version 1.24.0) [56]. Taxa that were not observed in more than 0.5% of the samples were excluded from the DESeq2 analysis.

### 4.11. Quantification and Statistical Analysis

All statistical analyses are reported as the mean ± SEM, and the differences in the relative abundance of bacterial populations in feces were analyzed using the Mann–Whitney sum rank tests in R software (R 4.2.0, RStudio, PBC, Boston, MA, USA). Significance was declared at *p* < 0.05, with Benjamini and Hochberg’s adjustment. All graphs were created using R software.

### 4.12. Ethics Approval

All the experimental procedures complied with the ARRIVE guidelines, and the Institutional Animal Care and Use Committee of the Korea Basic Science Institute approved the animal protocols (KBSI-IACUC-23-12).

## Figures and Tables

**Figure 1 ijms-25-00662-f001:**
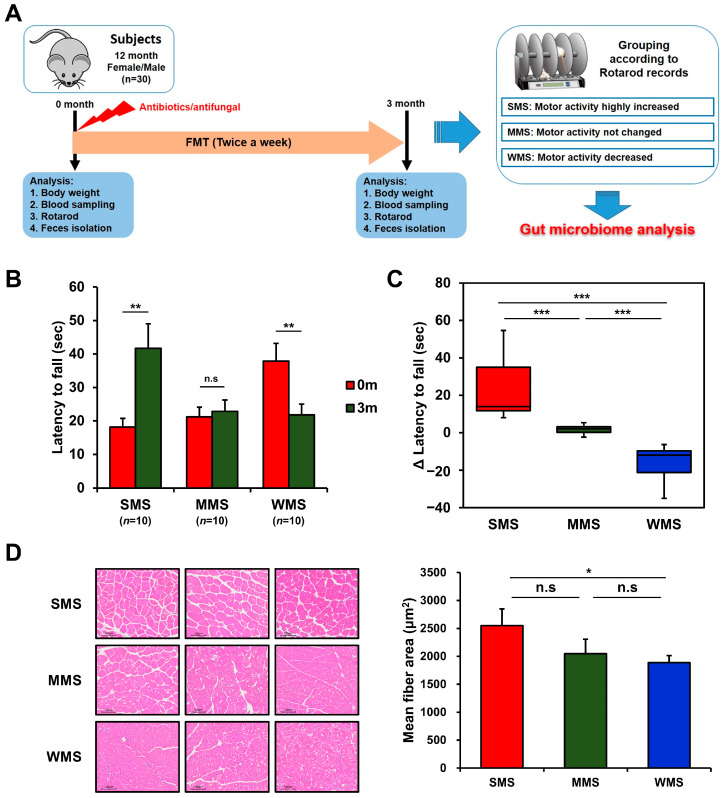
Grouping of mice based on changes in muscle strength before and after FMT. (**A**) A schematic diagram of the experimental design. (**B**) The comparison of rotarod performance of the experimental mice before and after FMT. (**C**) Changes in muscle strength of the experimental mice after FMT. (**D**) Representative histological images of H and E-stained muscle tissue (scale bar = 100 μm) from the mice with strong, medium, and weak muscle strength (SMS, MMS, and WMS, respectively). The values in the figure are expressed as the mean ± standard error of the mean (SEM) and the level of significance is indicated by asterisks (*): * *p* < 0.05, ** *p* < 0.01; *** *p* < 0.001; “n.s.” stands for “not significant” and indicates a *p*-value greater than 0.05.

**Figure 2 ijms-25-00662-f002:**
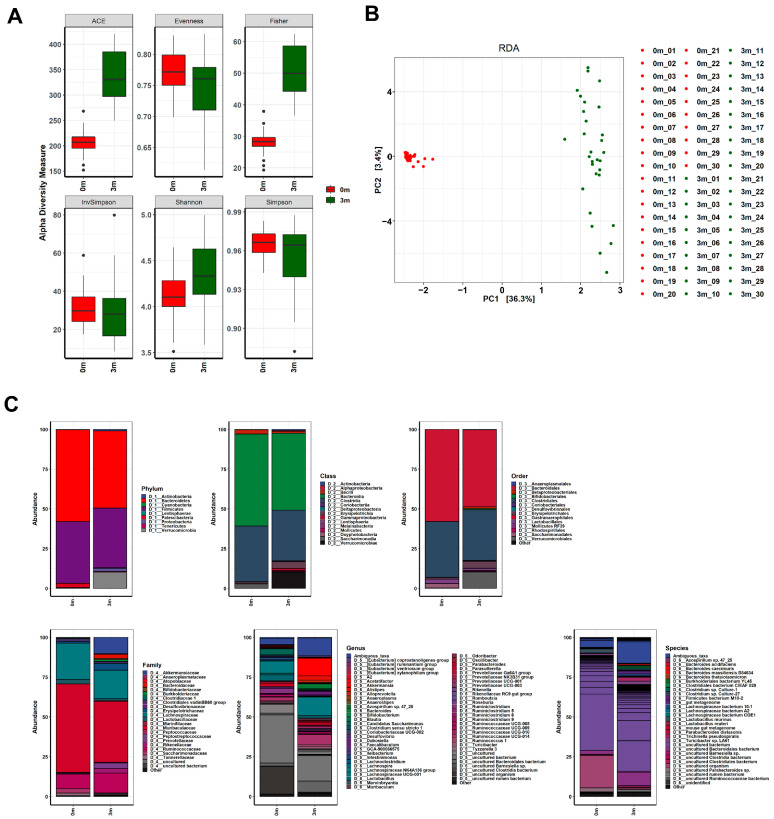
Changes in the gut microbiome before and after FMT. (**A**) The α-diversity, (**B**) β-diversity, and (**C**) abundance bar plot analyses revealed that the composition of the gut microbiome in the mice was significantly altered following FMT. Before the FMT, 0 m represents the baseline, while 3 m represents 3 months after the FMT.

**Figure 3 ijms-25-00662-f003:**
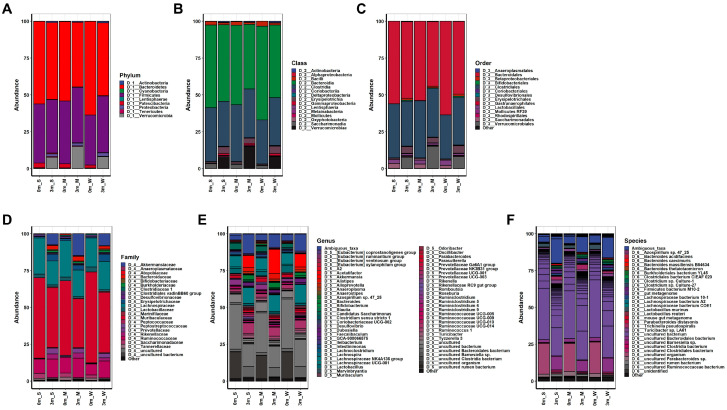
The composition of the gut microbiome in experimental mice before and after FMT. The relative changes in the gut microbiome composition were analyzed at the (**A**) phylum, (**B**) class, (**C**) order, (**D**) family, (**E**) genus, and (**F**) species levels. The average abundance values of each group were calculated to obtain their respective abundance values. S, M, and W represent the SMS group, MMS group, and WMS group, respectively.

**Figure 4 ijms-25-00662-f004:**
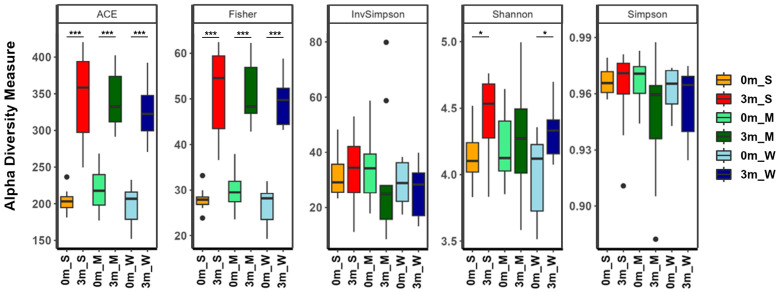
The α-diversity indices of the gut microbiome of the SMS, MMS, and WMS groups. The species richness and diversity, as calculated using ACE richness, Fisher’s alpha, inverse Simpson, Shannon diversity, and Simpson, are shown for both before (0 m) and after (3 m) FMT for each of the SMS, MMS, and WMS groups. S, M, and W represent the SMS group, MMS group, and WMS group, respectively. * *p* < 0.05, *** *p* < 0.001.

**Figure 5 ijms-25-00662-f005:**
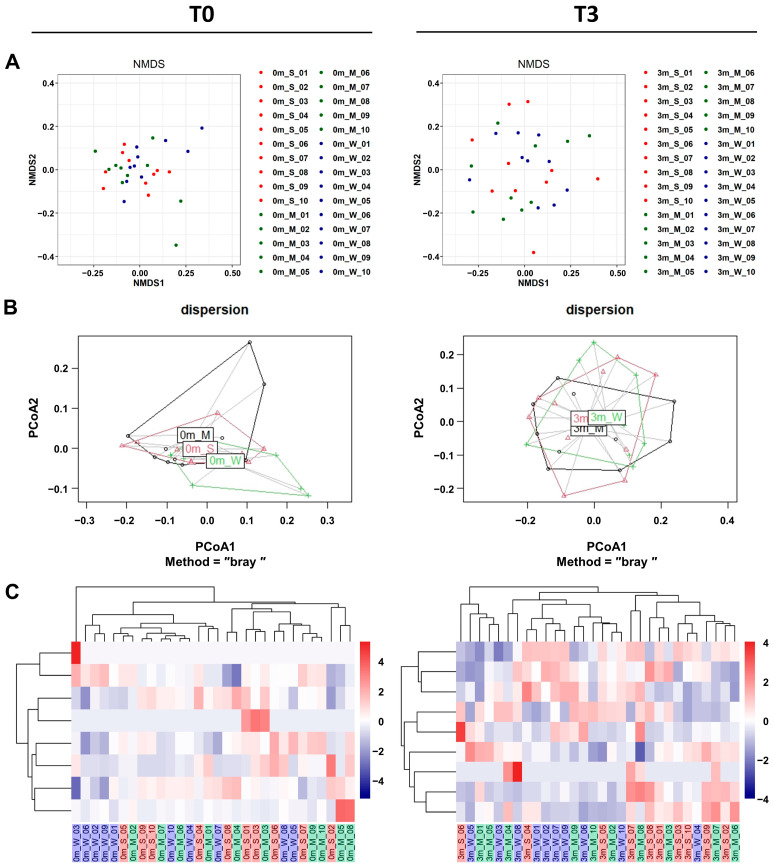
The β-diversity comparison of the gut microbiome of the SMS, MMS, and WMS groups. (**A**) Non-metric multidimensional scaling (NMDS) plots are shown, depicting the differences in the gut microbiome before (T0, left) and after (T3, right) FMT in the SMS, MMS, and WMS groups, based on Bray–Curtis distances calculated using operational taxonomic units (OTUs). (**B**) Principal coordinates analysis (PCoA) based on the Bray–Curtis distance from the PERMANOVA analysis (betadisper function) in the gut microbiome before (T0, left) and after (T3, right) FMT in the SMS, MMS, and WMS groups. (**C**) Heatmaps of the microbial composition of the SMS, MMS, and WMS groups before (T0, left) and after (T3, right) FMT are shown, based on the Bray–Curtis distance matrix measured at the phylum level. S, M, and W represent the SMS group, MMS group, and WMS group, respectively.

**Figure 6 ijms-25-00662-f006:**
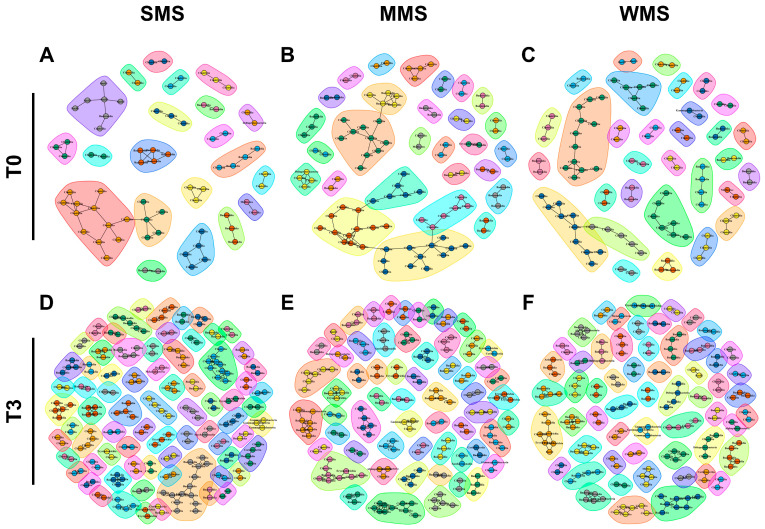
The co-occurrence network analysis using the ReBoot algorithm for the SMS, MMS, and WMS groups. The color-coded network graphs indicate the co-occurring and mutual exclusion interactions between operational taxonomic units (OTUs). Black letters in the nodes correspond to the class level of the OTUs. Transparent shapes represent network communities determined via the Louvain modularity algorithm. (**A**,**B**) The SMS groups before FMT (T0) and after FMT(T3), respectively. (**C**,**D**) The MMS groups before FMT (T0) and after FMT(T3), respectively. (**E**,**F**) The WMS groups before FMT (T0) and after FMT(T3), respectively.

**Figure 7 ijms-25-00662-f007:**
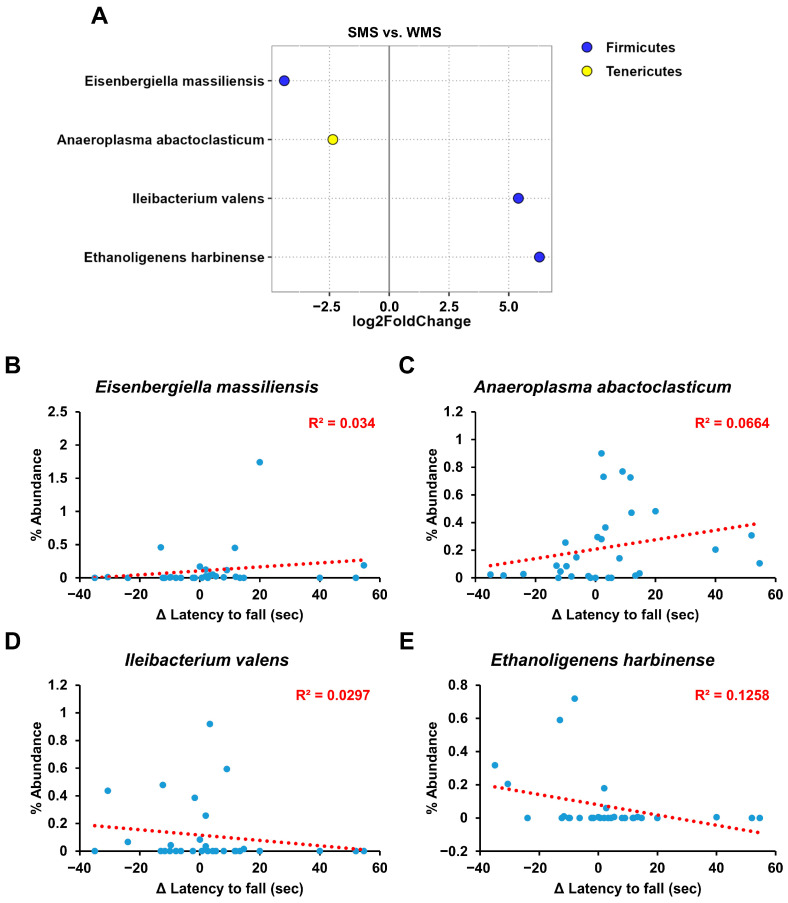
The key taxa changes between SMS and WMS via differential abundance analysis. (**A**) The log2-fold change in abundance of the most abundant species in the gut microbiome of the SMS and WMS groups was analyzed using DESeq2 differential abundance analysis. Each point represents a comparison of species between the two experimental groups. (**B**–**E**) The normalized abundances of four significantly different bacterial species of interest that were identified from the differential abundance analyses are shown. Scatter plots represent the % abundances of each individual species.

## Data Availability

The taxonomy profile data were deposited in the repository at figshare (https://doi.org/10.6084/m9.figshare.21838947) accessed on 9 January 2023.

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
