# Peer review of "Identification of Muscle Strength-Related Gut Microbes through Human Fecal Microbiome Transplantation"

_ijms, 2024, doi:10.3390/ijms25010662_

Round 1
Reviewer 1 Report
Comments and Suggestions for Authors
The manuscript presented by the authors determines the influence of gut microbes on muscle strength through fecal microbiome transplantation.
The quality of the work is outstanding, however, it is necessary to clear some aspects of the manuscript:
1)Abbreviations should be avoided in the part of the keyword.
2)Could the authors please explain why twice a week for a total of three months by administering 20 ul of the mixture was performed? I mean whether fecal microbiota transplantation cycles, frequency, and dosage will affect the recovery or growth of gut microbiota.
3)The manuscript is short of a "Conclusion".
4)Please list gut microbiomes that have a negative effect or positive effects on muscle strength by previous studies in the part of the discussion.
Comments on the Quality of English LanguageMinor editing of English language required
Author Response
- Abbreviations should be avoided in the part of the keywords.
Response to Comment:
As you advised, the abbreviated keyword "FMT" was deleted. Thank you.
- Could the authors please explain why twice a week for a total of three months by administering 20 ul of the mixture was performed? I mean whether fecal microbiota transplantation cycles, frequency, and dosage will affect the recovery or growth of gut microbiota.
Response to Comment:
Our purpose of FMT was not to ensure that the mice had intestinal microorganisms that completely matched the transplanted human intestinal microorganisms, but we performed FMT with the expectation that the human intestinal microorganisms accepted and established in each mouse would be different. In other words, we expected that the abundance of human intestinal microorganisms in each mouse would be different. It was expected that a too frequent FMT frequency would reduce the diversity of intestinal microorganism abundance, and a too short FMT period of time was expected to result in restoration to the mouse's own intestinal microorganisms, so a method of twice a week, for a total of 3 months, was selected. In terms of dosage, since fresh feces mixture was used, we decided to ingest small amounts over a long period of time to avoid possible disease outbreaks. Thank you.
- The manuscript is short of a “Conclusion”.
Response to Comment:
In line 324~327, we add the conclusion. Thank you.
- Please list gut microbiomes that have a negative effect or positive effects on muscle strength by previous studies in the part of the discussion.
Response to Comment:
In line 283~290, we added what you requested. Thank you.
Reviewer 2 Report
Comments and Suggestions for Authors
In this study, Ahn et al. identified some gut bacteria associated with muscle strength by comparing changes in gut bacteria in the same mouse before and after human fecal microbiota transplantation (FMT). After carefully reviewing this paper, I recognized there were a few issues. Here are some comments on this paper:
1. Figure 1, it is recommended that authors provide FMT frequency information in figure 1 A and add the numbers of mice in each group to figure 1 B.
2. As the author states in lines 121-122 “The effect of FMT on muscle strength in conventional mice was vastly different between mice, as shown in Figure 1B,C, probably due to the human gut microbiota randomly colonizing the mice gut.” Was it just a coincidence that 10 mice per group? Could the authors give any explanation?
3. It is proposed to replace figure 2 with figure S2.
4. Significant differences should be labeled in Figure 3.
5. Lines 441-444 “To 441 detect statistical differences in beta diversity metrics between groups, we used permuta- 442 tional multivariate analysis of variance (PERMANOVA) in the vegan package in R. ADO- 443 NIS was used with 999 permutations in the vegan package in R to quantify the effect size 444 of variables explaining the Bray-Curtis distance.” Where were the PERMANOVA results, are they shown in Figure 4?
6. Figure 4 C, grouping information or labels should be added.
7. Except for the phylogenetic tree, the content of the presentations in Figures 5 and 2 were the same, both relative abundance of taxa. It's hard to read the words in the phylogenetic tree. Proposed deletion of figure 5.
8. It's hard to read the words in Figure 6, could it be labeled in another way?
9. I consider rotarod performance scoring and gut microbiome relative abundance is a reasonable way to establish a correlation. For Figure 7, the correlation of the 3 points is not reasonable in my opinion, there is no way to calculate R2 and p values.
10. Line 354, how old were the mice when they were purchased?
11. I think the major flaw in this study is the lack of control groups, without the blank and negative controls. How did the muscle strength of untreated mice change over the 3-month experimental period? What is more, did the optimized media affect the mice? I think these are very important to this study.
12. Lines 413 and 415 “Metagenome sequencing” should be 16S rRNA gene sequencing.
13. Line 429, it was the result, not the raw sequencing data.
14. Line 438 “table.qzv”, the results were in qza file.
Author Response
- Figure 1, it is recommended that authors provide FMT frequency information in figure 1 A and add the numbers of mice in each group to figure 1 B.
Response to Comment:
As you advised, we provided FMT frequency information (twice a week) in Figure 1A, and added the number of mice in each group (n=10) in Figure 1B. Thank you.
- As the author states in lines 121-122 “The effect of FMT on muscle strength in conventional mice was vastly different between mice, as shown in Figure 1B,C, probably due to the human gut microbiota randomly colonizing the mice gut.” Was it just a coincidence that 10 mice per group? Could the authors give any explanation?
Response to Comment:
In order to equalize the number of mice in each group, after listing them in order based on the degree of change in wire suspension test records, they were divided into 10 mice with the greatest increase in muscle strength (SMS), 10 mice with no change in muscle strength (MMS), and 10 mice with decreased muscle strength (WMS). Therefore, there may not be much difference in the records of mice at the border of each group. However, it is a coincidence that the averages of the three groups are precisely divided into positive, non-change, and negative. Thank you.
- It is proposed to replace figure 2 with figure S2.
Response to Comment:
As you suggested, Figure S2 was replaced with Figure 2. Thank you.
- Significant differences should be labeled in Figure 3.
Response to Comment:
As you advised, we expressed significant differences as p-values. Thank you.
- Lines 441-444, “To detect statistical differences in beta diversity metrics between groups, we used permutational multivariate analysis of variance (PERMANOVA) in the vegan package in R. ADONIS was used with 999 permutations in the vegan package in R to quantify the effect size of variables explaining the Bray-Curtis distance.” Where were the PERMANOVA results, are they shown in Figure 4?
Response to Comment:
We added the PERMANOVA data and result table in Figure 5B and Table S9. Thank you.
- Figure 4 C, grouping information or labels should be added.
Response to Comment:
The group name was written as S, M, and W in the sample name. They were shaded in color for easy identification. (Red for SMS group, green for MMS group, blue for WMS group) Thank you.
- Except for the phylogenetic tree, the content of the presentations in Figures 5 and 2 were the same, both relative abundance of taxa. It’s hard to read the words in the phylogenetic tree. Proposed deletion of figure 5.
Response to Comment:
Original Figure 5 has been deleted. Due to the semantic similarity between the original Figure 2 and the original Figure 5, it was suggested that Figure 5 be deleted, so the original Figure 2 was retained as Figure 3. Thank you.
- It’s hard to read the words in Figure 6, could it be labeled in another way?
Response to Comment:
We reduced the size of the text in the network to avoid overlapping. Thank you.
- I consider rotarod performance scoring and gut microbiome relative abundance is a reasonable way to establish a correlation. For Figure 7, the correlation of the 3 points is not reasonable in my opinion, there is no way to calculate R2 and p-values.
Response to Comment:
As you recommend, to reasonably show the relationship between rotarod performance scores and the relative abundance of the gut microbiome, we plotted scatter plots and trend lines with each record from every individual instead of the average for each group. R2 and p-value were calculated through linear regression analysis. Thank you.
- Line 354, how old were the mice when they were purchased?
Response to Comment:
We purchased 47-week-old mice and acclimatized them for one week. Thank you.
- I think the major flaw in this study is the lack of control groups, without the blank and negative controls. How did the muscle strength of untreated mice change over the 3-month experimental period? What is more, did the optimized media affect the mice? I think these are very important to this study.
Response to Comment:
Since the purpose of this paper is to study the microbiome that influences muscle strength improvement through FMT of human intestinal microorganisms, a control group that did not undergo FMT was not necessary. Optimized media is a simple medium that allows anaerobic bacteria to grow well, and consuming only the medium did not change any of the physiological functions of the mouse. Thank you.
- Lines 413 and 415 “Metagenome sequencing” should be 16S rRNA gene sequencing.
Response to Comment:
As you advised, we replaced the word "Metagenome sequencing" to "16S rRNA gene sequencing". Thank you.
- Line 429, it was the results, not the raw sequencing data.
Response to Comment:
We changed the ‘raw data’ to ‘taxonomy profile data’. Thank you.
- Line 438, “table.qzv”, the results were in qza file.
Response to Comment:
As you pointed out, 'qzv' was changed to 'qza'. Thank you.
Reviewer 3 Report
Comments and Suggestions for Authors
Dear Editor,
Thank you for this opportunity to examine the following manuscript entitled ‘’ Identification of muscle strength-related gut microbes through human fecal microbiome transplantation ‘’ submitted for possible publication in the International Journal of Molecular Sciences. After carefully reviewing the abovementioned article, I found it interesting for publication. There are some issues to be addressed, accounting:
- The abstract section should be reworded and mention the main obtained results
- The introduction part is very lengthy and should be brief and concise
- Why did the authors adopt the combination of three antibiotics and one antifungal?
Author Response
- The abstract section should be reworded and mention the main obtained results.
Response to Comment:
As you advised, we rewrote the abstract mentioning the main results. Thank you.
- The introduction part is very lengthy and should be brief and concise.
Response to Comment:
We made the introduction concise by removing and condensing some sentences. Thank you.
- Why did the authors adopt the combination of three antibiotics and one antifungal?
Response to Comment:
Referring to previous FMT papers, we used the three most common antibiotics and one antifungal agent considering the intestinal environment. To verify this combination of antibiotics, we conducted a lot of research, and through several papers published by our research group, we can confirm that this combination is the optimal FMT system established through various trials and errors. Thank you.
Round 2
Reviewer 2 Report
Comments and Suggestions for Authors
Thank you for the authors' reply. My main points and concerns have been satisfactorily addressed. I have no further concerns.